# Online Learned Continual Compression with Stacked Quantization Modules

## Abstract

We introduce and study the problem of Online Continual Compression, where one attempts to learn to compress and store a representative dataset from a non i.i.d data stream, while only observing each sample once. This problem is highly relevant for downstream online continual learning tasks, as well as standard learning methods under resource constrained data collection. To address this we propose a new architecture which Stacks Quantization Modules (SQM), consisting of a series of discrete autoencoders, each equipped with their own memory. Every added module is trained to reconstruct the latent space of the previous module using fewer bits, allowing the learned representation to become more compact as training progresses. This modularity has several advantages: 1) moderate compressions are quickly available early in training, which is crucial for remembering the early tasks, 2) as more data needs to be stored, earlier data becomes more compressed, freeing memory, 3) unlike previous methods, our approach *does not require pretraining*, even on challenging datasets. We show several potential applications of this method. We first replace the episodic memory used in Experience Replay with SQM, leading to significant gains on standard continual learning benchmarks using a fixed memory budget. We then apply our method to online compression of larger images like those from Imagenet, and show that it is also effective with other modalities, such as LiDAR data.

## 1 Introduction

Interest in machine learning in recent years has been fueled by the plethora of data being generated on a regular basis. Effectively storing and using this data is critical for many applications, especially those involving continual learning. In general, compression techniques can greatly improve data storage capacity, and, if done well, reduce the memory and compute usage in downstream machine learning tasks (Gueguen et al., 2018; Oyallon et al., 2018). Thus, learned compression has become a topic of great interest (Theis et al., 2017; Ballé et al., 2016; Johnston et al., 2018). Yet its application in reducing the size of datasets bound for machine learning applications has been limited.

This work focuses on the following familiar setting: new training data arrives continuously for a learning algorithm to exploit, however this data might not be iid, and furthermore there is insufficient storage capacity to preserve all the data uncompressed. We may want to train classifiers, reinforcement learning policies, or other models continuously from this data, or use samples randomly drawn from it at a later point for a downstream task. For example, an autonomous vehicle (with bounded memory) collects large amounts of high-dimensional training data (video, 3D lidar) in a non-stationary environment (e.g. changing traffic patterns), and overtime applies an ML algorithm to improve its behavior using this data. This data might be transferred at a later point for use in downstream supervised learning. Current standard learned compression algorithms, e.g. Torfason et al. (2018), are not well designed to deal with this case, as their convergence speed is too slow to be usable in an online setting.

In the field of continual/lifelong learning (Thrun & Mitchell, 1995), which has for now largely focused on classification, approaches based on storing memories for later use have emerged as some of the most effective in online settings (Lopez-Paz et al., 2017; Aljundi et al., 2018; Chaudhry et al.; 2019; Aljundi et al., 2019). These memories can be stored as is, or via a generative model (Shin et al., 2017). Then, they can either be used for rehearsal (Chaudhry et al., 2019; Aljundi et al., 2019) or for constrained optimization (Lopez-Paz et al., 2017; Chaudhry et al.; Aljundi et al., 2018).

Indeed many continual learning applications would be nearly solved with replay approaches if one could afford to store all samples. These approaches are however inherently limited by the amount of data that can be stored[1].

Learning a generative model to compress the previous data stream thus seems like an appealing idea. However, learning generative models, particularly in the online (possibly non-stationary) setting, continues to be challenging, and can greatly increase the complexity of the continual learning task. Furthermore, such models are susceptible to catastrophic forgetting Aljundi et al. (2019). An alternate approach is to simply learn a compressed representation of the data; this is typically faster and more stable than learning to generate the whole data distribution. While the learned compression may itself exhibit forgetting and representation drift, causing challenges for continual and online cases, a learned compression method that can learn continuously and online would allow the storing of far larger amount of samples for replay.

In this work we investigate the use of quantized autoencoders, specifically the VQ-VAE framework (van den Oord et al., 2017), observing that these can learn continuously and online with minimal forgetting, particularly when augmented with their own internal rehearsal mechanisms. We propose a multi-level stacked model that allows the compressor to adaptively store samples at different compression scales, based on the amount of data, storage capacity, and effectiveness of the model in compressing samples. Furthermore, the learned compressed representation allows multiple continual learning models to be trained from the same data.

The main contributions in this work are as follows: (a) we introduce and highlight the importance of the online continual learned compression problem; (b) we demonstrate how Multi-level VQ-VAE, combined with internal replay, can effectively learn compressed representations of online data, (c) we show online learned compression can yield state-of-the-art performance in standard online continual image classification benchmarks.

## 2  RELATED WORK

**Learned compression**  has been recently studied for the specific case of image compression. Work by Theis et al. (2017); Ballé et al. (2016); Johnston et al. (2018) have shown learned compressions can outperform standard compression algorithms like JPEG. Some of these methods however are challenging to train, thus in our work we focus on the VQ-VAE approach van den Oord et al. (2017); Razavi et al. (2019) which allows us to address online continual learning settings and permit a multi-level storage.

**Continual Learning**  research currently focuses on overcoming catastrophic forgetting (CF) in the supervised learning setting, with some limited work in the generative modeling and reinforcement learning settings. Most continual learning methods can be grouped into three major families.

Some algorithms dynamically change the model's architecture to incorporate learning from each task separately. Popular methods are Rusu et al. (2016), Li & Hoiem (2018) and Fernando et al. (2017). Although these methods can perform well in practice, their introduction of task-specific weights requires growing compute and memory costs which are problematic for the online setting. Another set of techniques employ regularization to constrain weights updates in the hope of maintaining knowledge from previous tasks. Notable methods in this class include (Kirkpatrick et al., 2017; Huszár, 2017; Zenke et al., 2017; Nguyen et al., 2017; Chaudhry et al., 2018). This set of approaches is inefficient in the online setting Chaudhry et al. (2019).

The last family of methods encapsulates all that have a mechanism to store information about the previous data distributions. This *memory* then serves as a tool for the continual learner to rehearse previous tasks. The simplest instantiation of this method is to keep and sample from a buffer of old data to retrain the model after every update (Chaudhry et al., 2019). This approach is widely used in RL where it is known as Experience Replay (ER) (Lin, 1993). Another method, known as Generative Replay (GR) (Shin et al., 2017), uses generative modeling to store past tasks distributions. The continual learner then trains on generated samples to alleviate CF. Other notable examples are Gradient Episodic Memory (GEM) (Lopez-Paz et al., 2017), iCarl (Rebuffi et al., 2017), and Maximally Interfered Retrieval (MIR) Aljundi et al. (2019), as well as (Aljundi et al., 2018; Hu et al.,

---

[1]This is typically not feasible for on-device machine learning. And while many ML services currently run on the cloud, the move towards increasingly higher data privacy standards is likely to push many ML algorithms to run locally on device.

2018). Most closely related to our work Riemer et al. (2017) consider compressing memories for use in the continual classification task. They also employ a discrete latent variable model but with the Gumbel approximation which shows to be far less effective than our approach. Most notably a separate offline iid pre-training step for the learned compression is required in order to surpass the ER baseline, distinctly different from the online continual compression we consider.

**Lidar compression**   is considered in Tu et al. (2019) and Caccia et al. (2018). Both approaches use a similar projection from 3D $(x, y, z)$ coordinates to 2D cylindrical coordinates, and leverage deep generative models to compress the data. However, neither method was designed to account for potential distribution shift, nor for online learning. In this work we show that reusing this 2D projection in conjunction with out model allows us to mitigate the two issues above for lidar data.

---

**Algorithm 1:** SQM Replay

**Input:** Learning rates $\alpha_{cls}, \alpha_{ae}$
1 **Initialize:** Memory $\mathcal{M}$; $\theta_{ae}$
2 **for** $t \in 1..T$ **do**
3    % Fetch data from current task
4    **for** $B_{inc} \sim D_t$ **do**
5      **for** $n \in 1..N$ **do**
6        $B \leftarrow B_{inc}$
7        **if** $t > 1$ **then**
8          % Fetch data from buffer
9          $B_{re} \sim \text{SAMPLE}(M)$
10          $B \leftarrow (B_{inc}, B_{re})$
11        **end**
12        % Train Compressor Network
13        $\theta_{gen} \leftarrow ADAM(L_{vq}, B, \alpha_{ae})$
14        **if** $t > 1$ **then**
         UPDATEBUFFERREP$(M, \theta_{ae})$
15        %Save current indices
16        ADDTOMEMORY$(M, B_{inc}, \theta_{ae})$
17      **end**
18    **end**
19 **end**

---

**Algorithm 2:** AdaptiveCompress

**Input:** datapoint $x$, SQM with $L$ modules, distortion threshold $d_{th}$
1 **Initialize:** Memory $\mathcal{M}$; $\theta_{ae}$
2 % Compute hidden rep.  for all blocks
3 ENCODE$(SQM, x)$
4 % Iterate over blocks, from most compressed to least
5 **for** $i \in L..1$ **do**
6    % Fetch latent indices and hidden rep.  computed in forward pass
7    $\text{argmin}_i, z_q = \text{ARGMIN}(module_i)$
8    % Decode to output space
9    **for** $j \in i...1$ **do**    $z_q = \text{DECODE}(module_j, z_q)$
10    % Test reconstruction error
11    **if** $MSE(x, z_q) < d_{th}$ **then** **return** $\text{argmin}_i, i$
12 **end**
13 % If no compression is good enough, return original image
14 **return** $x, 0$

---

# 3 METHODOLOGY

In this section we outline our approach to the online continual compression problem. Our learned compression network consists of a set of Multi-resolution VQ-VAE blocks. These blocks only communicate information forward and are not learned jointly unlike the architecture presented in Razavi et al. (2019). Memories are compressed using an adaptive scheme that controls what resolution the sample is stored, and therefore how compressed the sample should be. Furthermore memories can be revisited and further compressed as the learned compression module improves. Finally a rehearsal phase that utilizes the stored memory is used to minimize forgetting and update representations stored in the memory.

## 3.1 PROBLEM SETTING

We consider the problem setting where a stream of samples $x \sim D_t$ arrives from different distributions $D_t$ changing over time $t = 1 \dots T$. We have a fixed storage capacity of $C$ bytes where we would like to store the most representative information from all data distributions $D_1, ...D_T$. There is notably a trade-off in quality of information versus samples stored. We propose to use a learned compression model, and most crucially, this model must also be stored within the $C$ bytes, to encode and decode the data samples. Another critical requirement is that at anytime $t$ the content of the storage (data and/or compression model) be usable for downstream applications. A key challenge is that the learned compression module will change over time, while we still need to be able to decode the memories in storage.

The high level training of the online learned compression is described in Algo 1. Random memories are decoded and used to train the current compression module, at the same time this also allows us to re-encode those memories using the update weights. The approach also incorporates a form of

replay to reduce forgetting and address *drift*, whereby the memories' quality degrade away from their original values. In the rest of this section we discuss the architecture, objective and storage we propose.

## 3.2 VECTOR QUANTIZED VAE

Variational Autoencoders (VAE) (Kingma & Welling, 2013) consist of two parts: the encoder network parameterizes the posterior distribution $q(z|x)$ and the decoder network $p(x|z)$ aims to reconstruct the original input $x$ from the inferred latent variables $z$. In a standard VAE the prior and posterior distributions are usually Gaussian with diagonal covariance. On the other hand Vector Quantized Autoencoders (VQ-VAE) use a discrete latent representation instead (van den Oord et al., 2017). This model additionally keeps an embedding table $E \in \mathbb{R}^{K \times D}$, consisting of $K$ vectors of size $D$. Given an input (e.g. an RGB image), the encoder first encodes it as a $H_h \times W_h \times D$ tensor, where $H_h$ and $W_h$ denote the height and width of the latent representation. Then, every $D$ dimensional vector goes through a discretization bottleneck using a nearest-neighbor lookup on the embedding table. Specifically, $z_{ij} = \arg\min_{e \in E} ||\texttt{enc}(x)_{ij} - e||_2$. The output of the discretization step is then fed through the decoder. The gradient of this non-differentiable step is approximated using the straight-through estimator. A key property to notice is that to reconstruct the input, only the $H_h \times W_h$ indices are required, thus yielding very powerful compression (van den Oord et al., 2017). The full VQ-VAE objective, $L_{vq}$ is given in van den Oord et al. (2017).

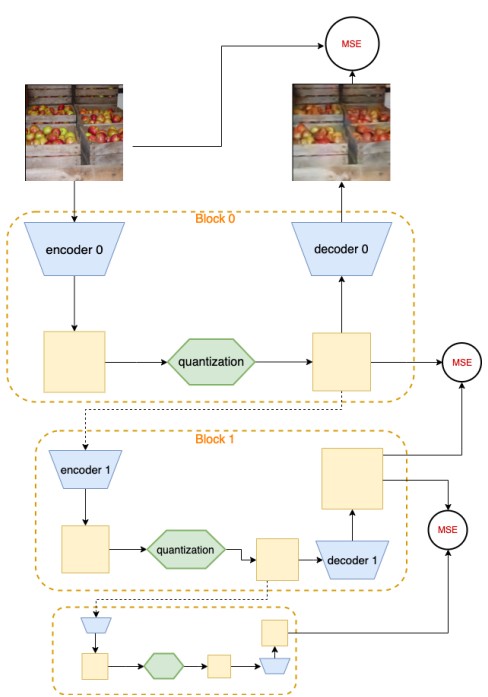

Figure 1: Stacked Quantization Modules. Each level uses its own loss and maintains its own replay buffer. Dotted lines indicate gradient is stopped in the backprop

## 3.3 STACKED QUANTIZATION MODULES

To ensure adaptivity of the compression model, we adopt a Stacked Quantization Modules (SQM). Each module contains a VQ-AE and a corresponding index buffer of adaptive capacity. Its input is the $z_q^{(i-1)}$ from the previous layer. A full diagram of the Stacked Quantization Modules (SQM) is given in Figure 1. Each module reconstructs its input from latent representation $z_q^i$, where $\text{BITS}(z_q^i) < \text{BITS}(z_q^{(i-1)})$. Here BITS denotes the memory in bits used by the encoding. The compression rate at a given block is given by

$$\frac{H \times W \times C \times \log_2(256)}{N_c \times H_{hi} \times W_{hi} \times \log_2(K_i)}$$

Thus the compression rate is controlled by: $K_i$, the number of embeddings in the codebooks of block $i$, the spatial dimension of the latent rep ($H_{hi}, W_{hi}$) and the number of codebooks $N_{ci}$.

VQVAE-2 (Razavi et al., 2019) also uses a multi-scale hierarchical organization, where unlike our SQM the top level models global information such as shape, while the bottom level, conditioned on the top one, models local information. While this architecture is tailored for generative modeling, it is less attractive for compression, as both the bottom and top quantized representations must be stored for high quality reconstructions.

Notably unlike VQVAE-2 (Razavi et al., 2019), each module is learned greedily without backward communication between modules using the current estimate of $z_q^{(i-1)}$ similar to Belilovsky et al.

(2019); Nøkland & Eidnes (2019). This formulation is important for allowing the modules to each converge as quickly as possible at their respective resolution. In other words a subsequent block is not required to build representations which account for all levels of compression, thus minimizing interference across resolutions. This rapid convergence is particularly important for the case of online continual learning.

## 3.4 MULTI-LEVEL STORAGE

Our aim is to store the maximum number of samples in the allotted $C$ bytes of storage, while assuring their quality, and our ability to reconstruct them. The SQM allows us to implement a Multi-Level storage, wherein each module stores samples at its respective scale. Samples are stored at different levels based on the compressors' current ability to compress them. When replay occurs, a sample may be able to propogate into a lower level (and thus permit more samples to enter the storage). This process is summarized in Algorithm 2. In practice, both the SQM training and the sample recompression are done in a single forward pass, allowing for fast training.

Such an approach is particularly helpful in the online continual learning setting and allows knowledge retention before the compressor network has learned a valid representations. Note that as per Algorithm 2, samples can be completely uncompressed until the first module is able to effectively encode them. This can be crucial in some cases, if the compressor has not yet converged, to avoid storing poorly compressed representations. Further taking into account that compression difficulty is not the same for all datapoints, this allows use of more capacity for harder data, and fewer for data.

We also note, since we maintain stored samples at each module and the modules are decoupled, that such an approach allows to distribute training the individual modules in parallel and in an asynchronous manner (Belilovsky et al., 2019).

## 3.5 MULTI-LEVEL RESERVOIR SAMPLING

Reservoir Sampling (RS) is a critical component of selecting a representative set of samples in continual classification Chaudhry et al. (2019). Its popularity is due to two reasons. First, it gives in expectation a sample representative of the whole data stream. Second, it is a conceptually simple algorithm, easy to implement, and gives strong empirical results. Indeed more sophisticated approaches can often be outperformed by basic RS in Chaudhry et al. (2019).

With this motivation, we need to adapt RS to a multi-level storage settings. There a few challenges. First, in our model the capacity in terms of samples is not fixed. RS adds a new point with prob. $p = \frac{\text{buffer capacity}}{\text{points seen so far}}$. However in practice, the buffer capacity actually *increases* as the compressor network gets better. We therefore estimate it using the current capacity at the time of buffer addition. In the same vein, one could argue that in practice the compressor network's sample capacity is actually bigger than the amount of samples it is currently storing since old samples were added when the buffer was less performant. However, since the points get re-compressed during rehearsal, this issue is mostly resolved.

Sample selection for deletion and sampling is also more complex than in standard RS. While it would be more advantageous to remove datapoints from the least compressed representation level, doing so introduces a bias; some classes may be harder to compress, and removing them more frequently would result in more forgetting of said class. To address this issue, we perform all buffer deletion (and insertion) agnostic to the level of the sample. When memory must be freed, samples are deleted from every block such that their relative memory consumption does not change.

## 3.6 PUTTING IT ALL TOGETHER - ONLINE COMPRESSION SQM

The online continual compression is overall summarized in Algorithm 1. Incoming data is added to memory as described in Algorithm 4 and along with randomly sampled data from storage is used to update the current SQM model. The randomly sampled data also updates its representation in storage as per Algorithm 3. Algorithm 2 describes the multilevel compression scheme. Note that Algorithm 1 can be run on its own or concurrently with a downstream application (e.g. online continual classification) which would use the same batch order as well as independently during a data collection phase.

---

**Algorithm 3:** UpdateBufferRep

---

**Input:** Memory $\mathcal{M}$, Autoencoder $AE$ with $L$ levels, data $D$, distortion threshold $d_{th}$
1 **for** $x \in D$ **do**
2      $hid_x, block_{id} = $ AdaptCompress$(x, AE, d_{th})$
3      % Delete Old Repr.
4      DELETE$(\mathcal{M}[x])$
5      % Add new one
6      ADD$(\mathcal{M}, hid_x)$
7 **end**

---

**Algorithm 4:** AddToMemory

---

**Input:** Memory $\mathcal{M}$ with capacity $C$ (bytes), sample $x$
1 $N_{reg} = \frac{C}{BYTES(x)}$
2 capacity = max( $N_{reg}$, NUM SAMPLES $(\mathcal{M})$ )

3 %Probability of adding x
4 add $\sim \mathcal{B}(\frac{\text{capacity}}{\text{SAMPLE AMT SEEN SO FAR}})$ %Bernoulli
5 **if** *add* **then**
6      $hid_x, block_{id} = $ AdaptCompress$(x, AE, d_{th})$
7      **while** *BITS($hid_x$) - FREE SPACE($\mathcal{M}$)* $> 0$ **do**
8          DELETE RANDOM$(\mathcal{M})$
9      **end**
10 **end**

---

## 4 EXPERIMENTS

We evaluate the efficacy of the proposed methods on a suite of canonical and new experiments. In Section 4.1 we present results on standard supervised continual learning benchmarks on CIFAR-10. In Section 4.2 we evaluate other downstream tasks such as standard iid training applied on the storage at the end of online continual compression. For this evaluation we consider larger images from Imagenet, as well as on lidar data.

### 4.1 ONLINE CONTINUAL CLASSIFICATION

Although CL has been studied in generative modeling (Ramapuram et al., 2017; Lesort et al., 2018; Zhai et al., 2019; Lesort et al., 2019) and reinforcement learning (Kirkpatrick et al., 2017; Fernando et al., 2017; Riemer et al., 2018), supervised learning is still the standard for evaluation of new methods. Thus, we focus on the online continual classification of images for which our approach can provide a complement to experience replay. In this setting, a new task consists of new image classes that the classifier must learn, while not forgetting the previous ones. The model is only allowed one pass through the data (Lopez-Paz et al., 2017; Chaudhry et al.; Aljundi et al., 2019; Chaudhry et al., 2019). The online compression here takes the role of replay buffer in replay based methods such as Chaudhry et al. (2019); Aljundi et al. (2019). In short, we apply Algorithm 1, with an additional online classifier being updated at line 13.

Here we consider the more challenging continual classification setting often referred to as using a *shared-head* (Aljundi et al., 2019; Farquhar & Gal, 2018; Aljundi et al., 2018). Here the model is not informed of the task (and thereby the subset of classes within it) at test time. This is in contrast to other (less realistic) CL classification scenarios where the task, and therefore subset of classes, is provided explicitly to the learner (Farquhar & Gal, 2018; Aljundi et al., 2019).

For this set of experiments, we primarily report accuracy, i.e. $\frac{1}{T}\sum_{i=1}^{T} R_{T,i}$, and forgetting, i.e. $\frac{1}{T-1}\sum_{i=1}^{T-1} \max(R_{:,i}) - R_{T,i}$ with $R \in \mathbb{R}^{T \times T}$ representing the accuracy matrix where $R_{i,j}$ is the test classification accuracy on task $j$ when task $i$ is completed.

**Baselines** A basic baseline for continual supervised learning is Experience Replay (**ER**). It consists of storing old data in a buffer to replay old memories. Due to its very simple nature, this baseline was often omitted in continual learning papers. However, recent research made it clear that it is a critical baseline to consider, and in some settings is actually state-of-the-art (Chaudhry et al., 2019; Aljundi et al., 2019; Rolnick et al., 2018). SQM can be viewed as an add-on to ER that incorporates online continual compression. In addition we consider the following baselines. **iid online** (upper-bound) trains the model with a single-pass through the data on the same set of samples, but sampled iid. **iid offline** (upper-bound) evaluates the model using multiple passes through the data, sampled iid. We use 5 epochs in all the experiments for this baseline. **fine-tuning** trains continuously upon arrival of new tasks without any forgetting avoidance strategy. **iCarl** (Rebuffi et al., 2017) incrementally classifies using a nearest neighbor algorithm, and prevents catastrophic forgetting by using an stored samples. **GEM** (Lopez-Paz et al., 2017) uses stored samples to avoid

| | Accuracy ($\uparrow$) | | Forgetting ($\downarrow$) | |
|---|---|---|---|---|
| | $M = 20$ | $M = 50$ | $M = 20$ | $M = 50$ |
| iid online | $60.8 \pm 1.0$ | $60.8 \pm 1.0$ | N/A | N/A |
| iid offline | $79.2 \pm 0.4$ | $79.2 \pm 0.4$ | N/A | N/A |
| GEM (Lopez-Paz et al., 2017) | $16.8 \pm 1.1$ | $17.1 \pm 1.0$ | $73.5 \pm 1.7$ | $70.7 \pm 4.5$ |
| iCarl (5 iter) (Rebuffi et al., 2017) | $28.6 \pm 1.2$ | $33.7 \pm 1.6$ | $49 \pm 2.4$ | $40.6 \pm 1.1$ |
| fine-tuning | $18.4 \pm 0.3$ | $18.4 \pm 0.3$ | $85.4 \pm 0.7$ | $85.4 \pm 0.7$ |
| ER | $27.5 \pm 1.2$ | $33.1 \pm 1.7$ | $50.5 \pm 2.4$ | $35.4 \pm 2.0$ |
| ER-MIR (Aljundi et al., 2019) | $29.8 \pm 1.1$ | $40.0 \pm 1.1$ | $50.2 \pm 2.0$ | $\mathbf{30.2 \pm 2.3}$ |
| (Riemer et al., 2018) | $25.5 \pm 2.0$ | $28.8 \pm 2.9$ | $71.5 \pm 2.8$ | $67.2 \pm 3.9$ |
| SQM (ours) | $\mathbf{39.9 \pm 0.8}$ | $\mathbf{46.2 \pm 0.8}$ | $50.5 \pm 1.1$ | $42.8 \pm 1.3$ |

Table 1: Shared head results on disjoint CIFAR-10. Total memory per class $M$ measured in sample memory size. We report (a) Accuracy, (b) Forgetting (lower is better).

increasing the loss on previous task through constrained optimization. It has been shown to be a strong baseline in the online setting. It gives similar results to the recent A-GEM Chaudhry et al.. **ER-MIR** (Aljundi et al., 2019) controls the sampling of the replays to bias sampling towards samples that will be forgotten. We note that the ER-MIR critera is orthogonal to SQM, and both can be applied jointly. We leave this as future work.

We evaluate with the standard CIFAR-10 split (Aljundi et al., 2018), where 5 tasks are presented sequentially, each adding two new classes. Evaluations are shown in Table 1. Due to our improved storage of previous data, we observe significant improvements over the other methods and baselines at various memory sizes. This is despite the drifting representation and decoder model. We can contrast SQM's performance with ER's to understand the net impact of our compression scheme. Specifically, SQM improves over ER by 12.4% and 13.1% in the M=20 and M=50 case, highlighting the effectiveness of online compression. Our approach only lags ER-MIR in forgetting in the M=50 setting. However, this method is orthogonal to ours and could thus be used jointly.

We also implemented the baseline of Riemer et al. (2018) that uses a discrete autoencoder with gumbel softmax (Jang et al., 2016). To compare directly to the other authors setup we benchmarked our implementation on the Incremental CIFAR-100 multi-head experiment Lopez-Paz et al. (2017). By using our (more sophisticated) architecture, we were able to get **60.3** vs the reported **43.7** using a buffer of size 200. While SQM performs similarly in this setting (**65.1**), its compressed representations incur significantly less drift compared to Riemer et al. (2018). In the single head setting, this distortion becomes much more problematic as the classifier learns to assign old labels to blurry images, leading to poor performance. We show an example of this drift in Fig. 4 in the Appendix.

The CIFAR-10 dataset has a low resolution ($3 \times 32 \times 32$) and uses a lot of data per task (10K samples). These two characteristics might leave the online compression problem easier than in a real-life scenario. Specifically, if the first tasks are long enough and the compression rate is not too large, the model can quickly converge and thus not incur too much representation drift. Indeed, we found that using a single module worked best for this task, as adding more modules was not worth the additional storage cost. For these reasons, we study the adaptive instantiation of our proposed method (A-SQM) in more challenging settings presented in the next section.

### 4.2 OFFLINE EVALUATION ON LARGER IMAGES AND LiDAR

Besides the standard continual classification setup, we propose several other evaluations to determine the effectiveness of the stored data and compression module after learning online compression.

**Offline training on Imagenet** We compare the effectiveness of the stored memories of SQM after a certain amount of online continual compression. We do this by training in a standard iid way an offline classification model using only reconstructions obtained from the storage sampled after online continual compression has progressed for a period of time. In each case we would have the same sized storage available. We remind the reader that simply having more stored memories does not amount to better performance as their quality may be severely degraded and affected by drift.

We use the mini-imagenet dataset, but resized to $128 \times 128$, larger than the typical size used as we aim to emphasize the utility of this methods for larger inputs. Online continual compression arrives

|                                          | Accuracy |
|------------------------------------------|----------|
| RS                                       | 5.2      |
| **2 Module A-SQM (ours)**                | **19.8** |
| Ablate Modular Training                  | 12.7     |
| Ablate Adaptive Compression              | 10.8     |
| Ablate 2nd Module                        | 9.61     |
| Ablate 2nd Module & Adaptive compression | 12.0     |

Table 2: Offline training evaluation of storage from online continual compression. We see a clear gain over a standard Reservoir sampling approach. We then ablate each component of our proposal showing each component is important. Note storage used in each experiment is identical (including accounting for model sizes).

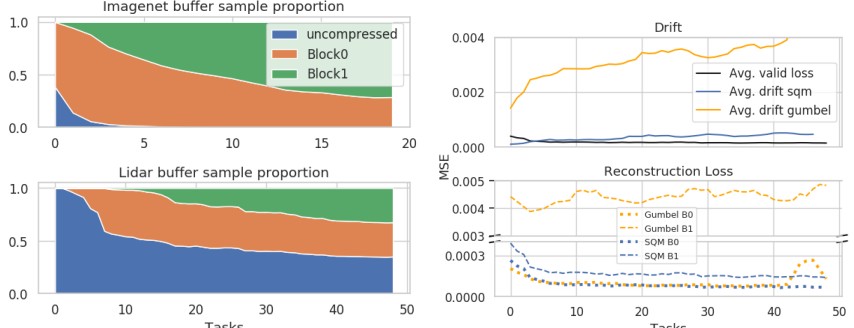

Figure 2: Left: Distribution of buffered samples for the Imagenet (top) and Lidar (bottom) experiments. We highlight that SQM's adaptive buffer is flexible enough to choose different sample proportions depending on the scenario. Right Top: Drift in the buffered lidar representations. Right Bottom: Blockwise validation loss for lidar evaluation

using the standard split-minimagenet from Chaudhry et al. (2019), which yields 20 different tasks and 100 classes total. After all samples have been seen and stored as best as possible in storage size $C$ by Algorithm 1, we train a Resnet18 model (similar to the one used in Chaudhry et al. (2019) adjusted for larger input size) using the stored samples. We train with SGD and a learning rate of 0.1, with early stopping using a validation set. The storage size $C$ is equivalent to 1000 uncompressed samples. Results of this evaluation are shown in Table 2.

Using this evaluation we first compare a standard reservoir sampling approach on uncompressed data to a 2 module A-SQM using the same size storage. We observe that performance is drastically increased using the compressed samples. We then use this to perform a series of ablations to demonstrate each component of our proposal is important. First we ablate the modular training, meaning the model is trained end-to-end instead of greedily, which is unable to perform well online. Indeed, we observe that the convergence speed is slower, resulting in fewer samples being stored for the early tasks. Secondly we consider not using the adaptive compression scheme described in Sec 3.4, thus all samples are compressed at the bottom level. This greatly decreases performance, for two reasons: the compressions stored early on are very poor since the model has not converged yet, and the model cannot allocate more capacity to more challenging samples.. We then consider removing the 2nd Module showing that multiple levels aid in performance. We additionally illustrate the change in the amount of samples stored at each level by adaptive storage in Figure 2.

**LiDAR** Range data can be very large and storage inefficient, it is also often collected by vehicles in potentially changing environments. Efficient storage can be important for having more representative data in downstream applications. Here we show qualitatively several examples of applying this. For this experiment we use the Kitti Dataset (Geiger et al., 2013), which contains 61 LiDAR scan recordings, each belonging to either the "residential", "road", "city" environments. The online compression is presented one by one with scans from each environment, we present all the recordings from one environment, before moving on to another. The data is processed as in Caccia et al. (2018), where points from the same elevation angle are sorted in increasing order of azimuth angle along the same row. This yield a 2D grid, making it compatible with the same architecture used in the previous experiments.

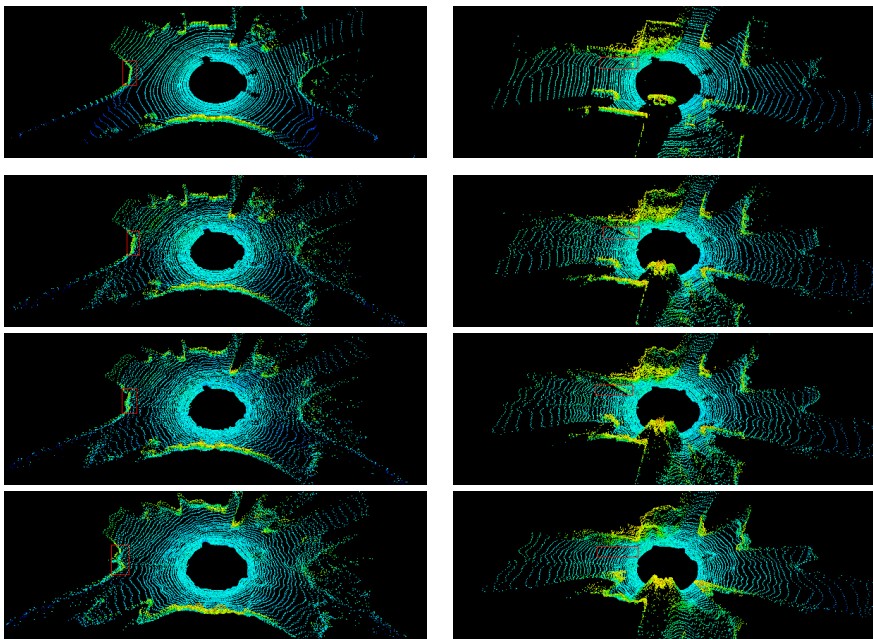

Figure 3: Top row are real uncompressed test set lidar scans. Rows two, three and four are reconstructions with compression rate 8, 16, and 32×. Regions of interest are highlighted in red. In the left column, we note all compression levels retain the essential information, albeit some far way obstacles are slightly deformed. In the right column, a thin obstacle is present in the first two levels, but absent in the third level.

We first show qualitative results in Figure 3, using a 3 Block SQM, observe that we are able to effectively reconstruct the LiDAR samples. and that we can easily tradeoff quality with compression. We note that reconstruction quality has been previously linked to performance on downstream tasks such as SLAM Zhang & Singh (2014). Note SQM can also be adapted to use task relevant criteria besides MSE(algo 2 line 9)

We now proceed with a quantitative evaluation of the LIDAR compression. We compare SQM with a 2-Block version of the Gumbel Autoencoder used in Riemer et al. (2017). We note that this baseline is stronger than in the original paper. We used our more sophisticated architecture, along with adaptive compression, which turned out to be essential to make the baseline competitive. In the challenging online continual compression setting, it is crucial that the stored representations remain valid as the model is updated. In Figure 2, We first look at representation drift incurred in the buffered samples. That is, after every task we decode every compressed representation and monitor the distortion with the true lidar scan (Fig. 2). The performance gap is quite stricking: SQM operates with drift values similar to the distortion observed at test time, throughout the whole 50 training task sequence. On the other hand, the representations learned from the Gumbel AE quickly stray away from their original and become unusable. We then analyse the stackability of blocks in SQM. For lidar datasets such as (Geiger et al., 2013), there is a high variance in the complexity of scans, therefore having multiple compression levels enables better storage allocation. In Fig 2, we observe that the difference in recontruction loss between blocks in SQM is small and well-behaved, which is not the case for the Gumbel baseline.

## 5 CONCLUSION

We have introduced online continual compression. We have shown how replay combined with a novel Multi-level quantization module can allow for the compression model to be learned online and yield a large decodable dataset despite representation and model drift. We have shown effectiveness of this online compression approach on standard continual classification benchmarks, as well as for compressing larger images and lidar data. We believe future work can consider dealing with temporal correlations for video and reinforcement learning tasks, as well as improved prioritization of samples for storage.

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

## A    FURTHER DETAILS OF EXPERIMENTS

We include here further details regarding the models used in the Imagenet and CIFAR-10 experiments. For all our experiments, we set D the size of the embedding table equal to 100. For CIFAR-10, we use a 1 block SQM, latent size (16 x 16 x 1) where the last index represents the number of codebooks. The codebook here contains 128 embeddings, giving a compression factor of 13.7. For Imagenet, we use a 2 block SQM. The first block has a latent size (32 x 32 x 2), and the second block has a latent size (32 x 32 x 1). All the codebooks have 128 embeddings, giving compression factors of 27.4 and 54.9 respectively.

## B    VISUAL COMPARISONS

Below we show an example of the image quality of our approach compared to Riemer et al. (2018).

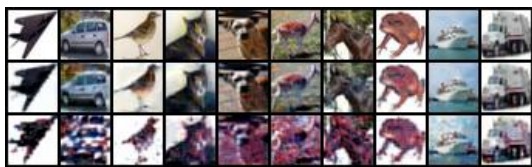

Figure 4: Bottom row: random buffer reconstructions using Riemer et al. (2018). Middle row: random buffer reconstructions using SQM. Top row: corresponding original image. Columns are ordered w.r.t their entry time in the buffer, from oldest to newest. All samples above were obtained from the disjoint CIFAR-10 task, and are 12× smaller than their original image.

