# OpenReview forum: "Online Learned Continual Compression with Stacked Quantization Modules"
_ICLR.cc/2020/Conference — Reject_

### Official Review · AnonReviewer3 · 2019-10-24
**Official Blind Review #3**

**Rating:** 6

**Review:**

This paper focuses on the problem of continual learning with limited memory storage. Specifically, the training data is arrived sequentially (might not be i.i.d.) for a model to exploit and there is not enough storage capacity to keep all the data without compression. This problem is important in many real-world applications with massive amount of data collected. The authors propose an approach named Stacked Quantization Modules to compress the data so that they can be stored efficiently. Each module is an auto-encoder with quantized latent representations. Several aspects including the communication between these stacked modules, and which level will a specific sample be compressed at, are taken into account in the algorithm design. In the experiments, the authors show some quantitative evaluations on CIFAR10 and ImageNet that the proposed method surpass several baseline methods. A qualitative visualization of LiDAR data reconstruction is also demonstrated. Overall I think the paper is tackling an interesting problem with an effective and novel solution.

I have a few concerns that I wish the authors could help to clarify. First, in the VQ-VAE, each image is quantized to be H*W*D, where each D-dimensional vector is represented by the index of the nearest neighbor in the embedding table of each module. I checked the paper but could not find a place that discuss how this embedding table comes from. It is pre-defined with some pattern or is it learnt somehow?

What is the latent space size of each module when trained on CIFAR10 and ImageNet?

The experiments on ImageNet only select 100 classes out of the 1000 classes. Would this method extends to large-scale datasets? How would the form of the tasks (in case of number of classes per task) affect the results?

There seems to be some typos. For example, the end of the first paragraph of Sec. 4.1 mentioned "line 13" of Alg. 4, which is not referred correctly as Alg. 4 only has 10 lines.

**Experience Assessment:**

I do not know much about this area.

**Review Assessment: Checking Correctness Of Derivations And Theory:**

I assessed the sensibility of the derivations and theory.

**Review Assessment: Checking Correctness Of Experiments:**

I assessed the sensibility of the experiments.

**Review Assessment: Thoroughness In Paper Reading:**

I read the paper at least twice and used my best judgement in assessing the paper.

---

> ### Author Response · Authors · 2019-11-14
> **Response to Reviewer**
>
> Thank you for your review. We respond to each point in turn.
>
> Re: the latent space embedding. It is learned as in the VQ-VAE, it is indeed critical to our applications that this works despite no pre-training. We also experiments with using a fixed codebook as in (https://arxiv.org/abs/1806.10474), but found the learning to be critical for fast convergence.
>
> Re: latent space size
> For all our experiments, we set D == 100. For cifar, we use a 1 block SQM,  latent size (16 x 16 x 1) where the last index represents the number of codebooks. The codebook here contains 128 embeddings, giving a compression factor of 13.7
> For Imagenet, we use a 2 block SQM. The first block has a latent size (32 x 32 x 2), and the second block has a latent size (32 x 32 x 1). All the codebooks have 128 embeddings, giving compression factors of 27.4 and 54.9 respectively. We have added this to the appendix
>
> Re: Imagenet class selection, we refer to the mini-imagenet dataset. We use the same setup as in the Chaudry et al paper. This is the most complex dataset considered in previous work for continual classification over long sequences. We also note for online continual classification in the shared-head setting (task id not available at test time) most existing methods fail (see e.g. https://arxiv.org/abs/1908.04742).  We do not see any reason our results would not extend to larger number of classes and longer task sequences. Indeed for continual classification SQM  combined with ER or ER-MIR should scale better than other method for longer sequences, as the representations and encoder/decoder learned online become more stable with longer data streams, and thus representational drift due to changing decoder becomes less of an issue.
>
> We have corrected many typos and generally revised the manuscript. Note that Algo 1 was mis-referenced as Algo 4, due to a typo in latex.

---

### Official Review · AnonReviewer1 · 2019-10-26
**Official Blind Review #1**

**Rating:** 3

**Review:**

This paper presented a Stacked Quantization Modules (SQM) for the problem of Online Continual Compression, based on the VQ-VAE framework by van den Oord et al. (2017). Experiments were conducted on online continual image classification benchmarks to show the effectiveness of the proposed SQM. In general, the novelty of the paper is a little bit limited and the writing of the paper is not very easy to follow.

- The SQM was constructed by stacking the known VQ-VAE. It is unclear why the stacking works for online continual compression. How many stacks should be used? What are the yellow rectangle parts in Figure 1?

- What are the relationship between Alg 1-4 ? More explanations or discussions are necessary.

- In Section 3.1, "The high level training of the online learned compression is described in Alg. 4.". It is very confused. I can't see the related content in Alg.4.

- In Section 4.1, "In short, we apply Algorithm 4, with an additional online classifier being updated at line 13." I don't understand it. I cannot see line 13 in Algorithm 4, because there is only 10 lines in Algorithm 4.

- In Section 3.3, BITS(.) needs definition.

- In Section 4.1, "Here we consider the more challenging shared-head setting, where the model is not informed of the task (and thereby the subset of classes) at test time. This is in contrast to other (less realistic) CL classification scenarios where the task, and therefore subset of classes, is provided explicitly to the learner Farquhar & Gal (2018)." It is very difficult to understand what the above experimental settings are.

- For Figure 3, the textures or lines of the bottom reconstructed one are not so smoothed or straight as the top one.

**Experience Assessment:**

I do not know much about this area.

**Review Assessment: Checking Correctness Of Derivations And Theory:**

I did not assess the derivations or theory.

**Review Assessment: Checking Correctness Of Experiments:**

I assessed the sensibility of the experiments.

**Review Assessment: Thoroughness In Paper Reading:**

I read the paper at least twice and used my best judgement in assessing the paper.

---

> ### Author Response · Authors · 2019-11-14
> **Response to Reviewer**
>
> Thank you for your review. We respond to each concern in turn below:
>
> Re:stacked modules and VQVAE novelty. The stacked modules are important for this task because it allows us to avoid compressing images to a maximum level until the learned compression is sufficiently effective on the particular task distribution (see also Sec 3.2 and abstract). At the same time their training without backward flow is efficient enough for online learning (Sec 3.3).  Furthermore we note that our work highlights that a single VQ-VAE can already be surprisingly effective at this problem. Although it is a known component, its development and application has been exclusively for the purpose of generative modeling, while its rapid convergence under non-stationary input has never been observed, studied, nor suggested before. Note other authors have unsuccessfully attempted to perform online continual compression to aid in specific downstream tasks (see e.g. Reimer et al. 2017 https://arxiv.org/abs/1711.06761). Finally the use of replay and updating indexes in the VQ-VAE training to aid forgetting in non-stationary settings is also quite different from prior works considering VQ-VAE’s.
>
> Re:Figure 1 Dashed lines.   Note the discussion at the end of Sec 3.3 the modules are learned independently without gradient flow between them (this is also supported empirically in the ablation studies of Table 2). The dashed lines thus indicate (as mentioned in the caption) where the gradient flow stops.
>
> Re: Shared-head setting. There is some history in the continual learning literature regarding this, several initial studies considered cases where the task is known at test time (e.g. for 100 possible image categories in https://arxiv.org/pdf/1706.08840.pdf) . This is a rather unrealistic setting and many authors have recently noted this and shifted focus to the so called “shared-head” or “single-head” setting (https://arxiv.org/abs/1805.09733, https://arxiv.org/pdf/1801.10112.pdf). This is important to note as otherwise it can cause confusion in comparing results across papers. We have added some references and another sentence to highlight more discussion of that must be found within those, we do believe further discussion into this aspect in the manuscript is outside the scope as this is becoming part of the standard evaluation protocol in continual classification.
>
> Re:“Relationship of Alg 1-4” We have added a new section to clarify this--Section 3.6, it discusses the relationships between each of the algorithms. We have also further refined the names of the algorithm blocks such that it is more explicit which algorithm blocks rely on other algorithm blocks
>
> Re: Algorithm 4 we apologize there was a referencing error in latex and it should refer to Algorithm 1.  This has been fixed
>
> Re: For the previous Fig 3, we note that while the edges of the reconstructions as not as smooth, the key components, such as cars and other obstacles, are fully visible and placed correctly.
>
> We added the definition of BITS
>
> Please let us know if there is further clarifications that can be made.

---

### Official Review · AnonReviewer2 · 2019-10-29
**Official Blind Review #2**

**Rating:** 6

**Review:**

The study tackled the problem of limited storage for ever-growing data for a long-term learning scenario. The authors proposed to stack Quantization Modules while separating them during training to obtain an online compression system that has multiple resolutions, different memory horizons, and reduced catastrophic forgetting. They also proposed a modified reservoir sampling to accommodate this architecture.

The idea is very simple yet interesting, the paper is a good read, and the results seem promising. The ablation study is well designed but not discussed enough. Additionally, the experiments cannot support the idea well since it is on a very special setting, the LiDAR experiment is missing quantitative evaluation, and different tasks (such as text classification, or visual tracking with only one labeled sample) might introduce different difficulties in this online learning setting. I recommend a weak accept for this paper to encourage the idea.

Therefore, I would recommend the authors to explore other tasks and see if their idea applies to different domains and tasks. Also, a quantitative evaluation for the LiDAR experiment with enough details and some explanation of the inner dynamics of the system during learning seems essential.

The paper could enjoy a pass of proofreading and typesetting (especially please pay attention to the correct use of \cite{} and \citep{}). Algorithm 1 is not mentioned in the body of the manuscript.

**Experience Assessment:**

I have read many papers in this area.

**Review Assessment: Checking Correctness Of Derivations And Theory:**

N/A

**Review Assessment: Checking Correctness Of Experiments:**

I carefully checked the experiments.

**Review Assessment: Thoroughness In Paper Reading:**

I read the paper thoroughly.

---

> ### Author Response · Authors · 2019-11-15
> **Response to Reviewer**
>
> Thanks for your time reviewing and help improving our paper!
>
> We agree that further experiments in settings besides the standard continual image classification settings would be valuable. We have thus now expanded the evaluation of the LIDAR adding experiments in Fig 2 and the last paragraph of Sec 4. We would like to note also that the offline imagenet evaluations performed in Sec 4.2 are a distinct application from those typically considered in the literature (e.g. those in Sec 4.1). Indeed the approach shows that non-iid data can be collected and compressed online and used in subsequent downstream applications at a later point.
>
> We believe this approach can also be very useful in applications in reinforcement learning, particularly ones that already rely on replay memory particularly ones with changing environments (e.g. Rolnick 2019). This however is beyond the scope of the current work
>
> Regarding the ablations we have extend the paragraph discussing this to give more insight. We have also corrected the typo you mentioned along with generally revising the text in the manuscript (see General comments for more details). Note the issue regarding Algorithm 1 reference: it was referenced as Algo 4. This has been corrected, see general comments.

---

### Official Review · AnonReviewer4 · 2019-11-29
**Official Blind Review #4**

**Rating:** 3

**Review:**

This work contributes to introducing a problem called Online Continual Compression. This problem requires to avoid catastrophic forgetting and learn in an online way. Generative methods should be one of the popular ways to do continual learning. This work’s model can be categorized into this clue since it also aims to save samples from old tasks by learning a generative model. In this way, the generator plays a similar role Experience Replay (ER) (here is called Generative Replay). The main core of this work should be the stacked quantization modules (SQM) which can be regarded as a hierarchical variant of the VQ-VAE model. In their SQM, hidden encodings z_q^i will be encoded and its input is z_q^{i-1} which is from previous layer.

This works covers related works very well. However, there are some questions I am really concerned:
1)	About the studied problem “Online Continual Compression”, what’s the difference between “online” and “continual”? In continual learning, tasks will be learned sequentially, right? If so, continual learning should run in an online learning way.
2)	The motivation of the hierarchy in this work is unclear. What I mean is that the hierarchical model should be expected to capture higher-level semantic features. But in this work, the index outputs z_q^{i-1} is encoded by its subsequent layer. It seems a bit weird since the z_q^{i-1} is not an image and its elements are index values. So what is the higher-level semantic information? By the way, it seems that there is an error in the model figure 1. The last MSE from Block 1 should be connected to the block before decoder 1 in Block 1, rather than the reconstructed one from decoder 1. Therefore, I strongly suggest authors give more insights and clarify the motivation of hierarchy. Writings in the METHODOLOGY part is unclear. More details about the SQM model should be described in a mathematical way.
3)	Another question about the details of generative replay. How do you do the replay? Details about this can’t be found in this work? In Alg.1, what is the \theta? Is the \theta_{ae} at line 14 of Alg.1 wrong? It should be \theta_{gen}, right?
4)	You use the data-stream technique reservoir sampling to add and update the memory buffer (alg. 4). Will it lead to some information loss? Can we just update memory without reservoir sampling? Please give more insights about this.
5)	How to find the distortion threshold d_th in Alg.2?
6)  the part of ablation studies is good. But I suggest authors should consider a baseline with the same proposed framework but using a single-layer VQVAE with the same memory capacity as the hierarchical models.


**Experience Assessment:**

I have read many papers in this area.

**Review Assessment: Checking Correctness Of Derivations And Theory:**

N/A

**Review Assessment: Checking Correctness Of Experiments:**

I assessed the sensibility of the experiments.

**Review Assessment: Thoroughness In Paper Reading:**

I read the paper at least twice and used my best judgement in assessing the paper.

---

### Official Review · AnonReviewer5 · 2019-12-05
**Official Blind Review #5**

**Rating:** 3

**Review:**

I am not familiar with the generative model and continual learning. Thus, I can only give my review based on the authors writing and other reviewers' comments.
- The paper proposes a new problem setup as "online continual compression".
- The paper gives a combination of many existing techniques to address the new problem. (I agree with Review #4)

I think the presentation and the organization of this paper should be improved in order to properly place their contributions in the literature.
- Since the authors try to promote a new problem set up with a solution containing little technical breakthroughs, I suggest the authors put more space on motivating the application and showing its impotence. Currently, their presentation focuses too much on methodology parts.
- Thus, it is better to organize the paper as an application from LiDAR and convince reviewers why their method is good for such an application (Review #1 also thinks the presentation is poor).
- If the authors insist on keeping their paper as a  methodology one, at least one more experiment (not the synthetic one on ImageNet) from real applications are needed (same as Review #2).

Overall, I think the paper is not ready for being published.

**Experience Assessment:**

I have read many papers in this area.

**Review Assessment: Checking Correctness Of Derivations And Theory:**

I assessed the sensibility of the derivations and theory.

**Review Assessment: Checking Correctness Of Experiments:**

N/A

**Review Assessment: Thoroughness In Paper Reading:**

I read the paper at least twice and used my best judgement in assessing the paper.

---

### Author Response · Authors · 2019-10-18
**Code Release**

Hi,

You can find the anonymized code to replicate our experiments here : https://github.com/StackedQuantizationModules/stacked-quantization-modules

---

### Author Response · Authors · 2019-11-15
**Manuscript Revisions**

Dear Reviewers, we thank you for your reviews that have helped us to revise the paper. We respond to each of your comments individually. Here we would like to highlight the new material in the paper and also note the main changes we have made to improve  the clarity in the manuscript:

-As noticed by all the reviewers, Algorithm 1 was misreferenced as Algorithm 4, this has been now corrected. Our latex file unfortunately had an error which caused this.
-We have added an additional section to further clarify the relation between all the algorithms (sec 3.6
- Through communication with the authors of “Scalable Recollections for Continual Lifelong Learning”  we have been able to reproduce their results on the Split-CIFAR100 task and include it in the paper a direct comparison (Parag 5 of sec 4.1), obtaining far better results than this related work.
-  We have revised the images for LIDAR in Fig 3 to display further images with highlighting of the key parts of the reconstruction. We have also added additional quantitative analysis of the LIDAR compression in Fig 2 and end of Sec 4.
- We have added additional analysis to illustrate how the distribution of samples stored at the different levels
- We have made all minor grammatical/spelling corrections noted by the reviewers.

---

### Decision · Program_Chairs · 2019-12-19

**Decision:**

Reject

**Comment:**

The paper proposes a new problem setup as "online continual compression". The proposed idea is a combination of existing techniques and very simple, though interesting. Parts of the algorithm are not clear, and the hierarchy is not well-motivated. Experimental results seem promising but not convincing enough, since it is on a very special setting, the LiDAR experiment is missing quantitative evaluation, and different tasks might introduce different difficulties in this online learning setting. The ablation study is well designed but not discussed enough.